# Prognostic Value of the De Ritis Ratio for Overall Survival in Patients with Metastatic Castration-Resistant Prostate Cancer Undergoing [^177^Lu]Lu-PSMA-617 Radioligand Therapy

**DOI:** 10.3390/cancers15204907

**Published:** 2023-10-10

**Authors:** Sebastian Gaal, Kai Huang, Julian M. M. Rogasch, Hans V. Jochens, Maria De Santis, Barbara Erber, Holger Amthauer

**Affiliations:** 1Department of Nuclear Medicine, Charité–Universitätsmedizin Berlin, Augustenburger Platz 1, 13353 Berlin, Germanyholger.amthauer@charite.de (H.A.); 2Praxen für Diagnostische und Therapeutische Nuklearmedizin, Düppelstr. 30, 12163 Berlin, Germany; 3Berlin Institute of Health, Charité–Universitätsmedizin Berlin, Charitéplatz 1, 10117 Berlin, Germany; 4Department of Urology, Charité–Universitätsmedizin Berlin, Charitéplatz 1, 10117 Berlin, Germany; 5Department of Urology, Medical University of Vienna, 1090 Vienna, Austria

**Keywords:** De Ritis ratio, prostate cancer, radioligand therapy, lutetium, PSMA

## Abstract

**Simple Summary:**

Radioligand therapy with [^177^Lu]Lu-PSMA-617 is an effective treatment for patients with prostate cancer. However, survival after radioligand therapy differs widely because patients respond differently to treatment but also enter therapy with an individual set of risk factors. We retrospectively analyzed the prognostic factors and survival data of 91 patients who were treated in our hospital. We found that patients with more previous lines of chemotherapy, higher levels of the prostate-specific antigen, and a higher “De Ritis ratio” (calculated from two laboratory parameters) lived—on average—for a shorter time after radioligand therapy than patients without these factors. We developed a score to better define patients with different survival outcomes after treatment. Between the highest and lowest risk groups, survival ranged from approximately 5 months to 28 months, respectively. If an independent validation of this score is successful, it could help doctors identify those patients not likely to benefit from radioligand therapy.

**Abstract:**

The De Ritis ratio (=aspartate transaminase/alanine transaminase) has shown prognostic value in different cancer types. This is the first such analysis in prostate cancer patients undergoing radioligand therapy (RLT) with [^177^Lu]Lu-PSMA-617. This retrospective monocentric analysis included 91 patients with a median of 3 RLT cycles (range 1–6) and median cumulative activity of 17.3 GBq. Univariable Cox regression regarding overall survival (OS) included age, different types of previous treatment, metastatic patterns and different laboratory parameters before RLT. Based on multivariable Cox regression, a prognostic score was derived. Seventy-two patients (79%) died (median follow-up in survivors: 19.8 months). A higher number of previous chemotherapy lines, the presence of liver metastases, brain metastases, a higher tumor load on PSMA-PET, a higher prostate-specific antigen (PSA) level, lower red blood cell count, lower hemoglobin, higher neutrophil-lymphocyte ratio and higher De Ritis ratio were associated with shorter OS (each *p* < 0.05). In multivariable Cox, a higher number of chemotherapy lines (range, 0–2; *p* = 0.036), brain metastases (*p* < 0.001), higher PSA (*p* = 0.004) and higher De Ritis ratio before RLT (hazard ratio, 1.27 per unit increase; *p* = 0.023) remained significant. This prognostic score separated five groups with a significantly different median OS ranging from 4.9 to 28.1 months (log-rank test, *p* < 0.001). If validated independently, the De Ritis ratio could enhance multifactorial models for OS after RLT.

## 1. Introduction

Radioligand therapy (RLT) with [^177^Lu]Lu-prostate-specific membrane antigen (PSMA) ligands is a palliative treatment option for patients with metastatic castration-resistant prostate cancer (mCRPC) [1]. In the VISION trial, RLT with [^177^Lu]Lu-PSMA-617 prolonged overall survival (OS) in patients with mCRPC compared to the standard of care such as the androgen receptor pathway inhibitor (median OS, 15.3 vs. 11.3 months) [2].

However, responses differed widely between patients. In the VISION trial, about 30% of patients achieved complete or partial response, while 10% of patients showed primary progressive disease [2]. Furthermore, RLT is currently offered mostly to patients who have undergone various other treatment lines, usually including at least one androgen receptor pathway inhibitor (abiraterone or enzalutamide) and often one or two lines of chemotherapy (docetaxel, cabazitaxel). Patients that are considered candidates for RLT, therefore, often suffer from end-stage disease with a high tumor burden, different degrees of myelosuppression, or impaired performance status, all of which have been identified as poor prognostic factors of shortened OS after RLT [3,4,5,6,7,8]. In general, mCRPC patients are offered RLT in different disease stages and with different numbers of pretreatments, which causes a considerable variation in such risk factors at the RLT baseline. It is, therefore, important to better define individual prognosis when discussing the indication, risks and benefits of RLT or best supportive care with the individual patient.

The prognostic value of the De Ritis ratio (the ratio of aspartate transaminase [AST] and alanine transaminase [ALT]) has not been studied in the context of RLT for patients with prostate cancer. However, a high De Ritis ratio has been linked to a poor prognosis in patients with localized prostate cancer [9] (optimal threshold, >1.325), in patients with mCRPC treated with Cabazitaxel [10] (threshold, ≥1.35), in other urological carcinomas [11,12,13,14] (threshold ranging from 1.26 to 1.5) and in patients with neuroendocrine tumors undergoing peptide receptor radionuclide therapy with [^177^Lu]Lu-DOTATOC [15] (threshold, >0.927).

The aim of this study was, therefore, to evaluate the independent prognostic value of the De Ritis ratio in light of other established prognostic factors regarding the OS of patients with mCRPC undergoing RLT with [^177^Lu]Lu-PSMA-617.

## 2. Materials and Methods

### 2.1. Patients

Using the hospital’s information system, we identified 134 patients with mCRPC treated consecutively with [^177^Lu]Lu-PSMA RLT between June 2015 and July 2020 at Charité–Universitätsmedizin Berlin. Patients with an insufficient follow-up duration (<6 months), incomplete follow-up information (lost to follow-up), activity per RLT cycle < 4.5 GBq (for risk of bias), deviation from the standard RLT regimen or lack of information on crucial prognostic parameters were excluded. A total of 91 patients were eligible for this monocentric, retrospective analysis (Figure 1).

These 91 patients also fulfilled the following inclusion criteria: (1) all vital tumor lesions were PSMA positive in positron emission tomography/computed tomography (PET/CT), (2) patients showed progressive disease following the last treatment before RLT. [^18^F]FDG PET was usually not performed in addition to the baseline PSMA PET.

### 2.2. [^177^Lu]Lu-PSMA RLT

RLT was performed with [^177^Lu]Lu-PSMA-617 at a standard prescribed activity of 6.0 GBq per cycle. The activity was adjusted in case of significantly reduced kidney function or significant myelosuppression. The standard interval between RLT cycles was 8 weeks. After the application of two cycles of RLT, all patients underwent PSMA PET/CT (usually [^68^Ga]Ga-PSMA-11) for restaging, and this was repeated every two cycles. In patients with progressive disease in PSMA PET/CT (i.e., new lesions or unequivocal progression of preexisting lesions in contrast-enhanced CT) or intolerable toxicity, treatment with RLT was discontinued. Furthermore, RLT ended after a maximum of six cycles. Patients were monitored with PSA after the last cycle of RLT, and imaging was used if required (i.e., in the case of suspected disease progression).

### 2.3. Baseline Parameters

Baseline parameters (i.e., before the first cycle of RLT) were retrieved from our hospital information system. These parameters included patient age, previous treatment lines, the pattern of organs affected by metastases at the start of RLT and laboratory parameters at the start of RLT (obtained within a few days before the first cycle). Laboratory parameters included PSA, plasma creatinine and the estimated glomerular filtration rate (eGFR) according to the Chronic Kidney Disease Epidemiology Collaboration (CKD-EPI), blood cell counts and hemoglobin, alkaline phosphatase, aspartate transaminase (AST) and alanine transaminase (ALT).

Normal values were as follows:Alkaline phosphatase: 40–130 U/L;White blood cell count: 3.9–10.5/nL;Red blood cell count: 4.3–5.8/pl (<65 years of age), 4.0–5.6/pl (>65 years);Hemoglobin: 13.5–17.0 g/dL (<65 years), 12.5–17.2 g/dL (>65 years);Platelet count: 150–370/nL;Neutrophil count: 1.5–7.7/nL;Lymphocyte count: 1.1–4.5/nL;Aspartate transaminase (AST): <50 U/L;Alanine transaminase (ALT): ≤41 U/L.

Using these values, we calculated the De Ritis ratio (=AST/ALT), the neutrophil–lymphocyte ratio (NLR = neutrophil count/lymphocyte count), and the platelet–neutrophil ratio (PNR = platelet count/neutrophil count).

Furthermore, the tumor load on the baseline PSMA-PET/CT was categorized into very low vs. low vs. moderate vs. high vs. very high tumor load following the example of Gafita et al. [16]. Gafita et al. used the semiautomated segmentation of PSMA-PET images to obtain the total tumor volume for each patient. Using the quintiles of total PSMA-positive tumor volume in their patients, they defined these five categories. We did not segment the tumor volumes (because the qPSMA software used by Gafita et al. is not openly accessible). In the current analysis, an experienced nuclear medicine physician (JMMR) estimated the tumor load visually. Representative patient examples for each category from the original publication [16] were used as a reference.

### 2.4. Follow-Up and Survival Status

Overall survival was defined as the time between the start of the first RLT cycle and death from any cause. Survival status was obtained based on data in the hospital’s clinical information system and tumor information system, by contacting the patient or the patient’s general practitioner or urologist/oncologist. Follow-up ended on the 7 August 2023. Approval for this analysis was given by the ethics commission of the Charité–Universitätsmedizin Berlin (vote, EA1/199/20), and patients gave their consent for data analysis and to obtain survival data upfront.

### 2.5. Statistical Analysis

Statistical analysis was performed using SPSS version 27 (IBM, Chicago, IL, USA) and R 4.1.3 (Foundation for Statistical Computing, Vienna, Austria, 2022). Significance was assumed at α = 0.05. Based on the Shapiro–Wilk test, non-normal data distribution for several variables was assumed, and descriptive values were generally expressed as the median, interquartile range (IQR) and range.

Univariable Cox proportional hazards regression regarding OS included patient age, previous lines of therapy, the pattern of metastases at the start of RLT, and laboratory parameters at the start of RLT. The hazard ratio (HR) and its 95% confidence interval (95% CI) were determined for each parameter. Parameters with a continuous scale were included as continuous variables. All variables with *p* ≤ 0.05 in univariable Cox regression and available data for all patients were candidates for multivariable Cox regression. The set of variables in the final model was identified with forward stepwise inclusion based on changes in the likelihood ratio (SPSS). Compliance with the proportional hazards assumption was tested using the goodness-of-fit test in the *survival* package for R and was fulfilled by all variables.

The predictive accuracy of the final Cox model was expressed using Harrell’s C (C index) calculated with the *rms* package in R. The *nomogram* function in the *rms* package was employed to generate a nomogram for the final multivariable Cox model. Predicted and observed survival probabilities were plotted with the *val.surv* function. Using equal weights, all final variables were also combined as a prognostic score that could also be used to predict OS. For this score, continuous variables in the final Cox model were binarized based on the lowest *p*-value in the log-rank test using the Cutoff Finder [17]. The Cutoff Finder was also used to generate a plot depicting HR and its 95% CI at various possible thresholds for the De Ritis ratio. The Kaplan–Meier method was used to illustrate survival curves and estimate survival duration.

Furthermore, patient groups with a low vs. high De Ritis ratio were compared using the Wilcoxon rank-sum test (continuous variables) or the two-sided Fisher’s exact test (categorical variables), respectively.

The association between certain baseline parameters (tumor load category, PSA, De Ritis ratio) was analyzed using Spearman’s rho and illustrated with box plots.

## 3. Results

### 3.1. Patients

Table 1 shows all patient characteristics. A total of 246 cycles of RLT were performed in 91 patients with a median of three cycles per patient (IQR, 2–3; range, 1–6). The median activity per cycle was 6.0 GBq (IQR, 5.7–6.0 GBq; range, 4.5–6.5 GBq), and the median cumulative activity of all cycles per patient was 17.3 GBq (IQR, 11.9–18.2 GBq; range, 4.5–35.9 GBq). In nine patients, only one cycle of RLT was performed because these patients died before the scheduled second cycle could be conducted.

Several baseline variables differed significantly between patients with a high or low De Ritis ratio, respectively (Table 1).

The baseline De Ritis ratio showed a weak but significant correlation with the PSA level (Spearman’s rho, 0.38; *p* < 0.001; Figure 2). The De Ritis ratio was also correlated with the tumor load on PSMA-PET (5 categories; Spearman’s rho, 0.45; *p* < 0.001). The baseline PSA level showed a moderate correlation with the tumor load on PSMA-PET (Spearman’s rho, 0.54; *p* < 0.001).

### 3.2. Overall Survival

During follow-up, 72 out of 91 patients (79%) died. The median OS in the total cohort of 91 patients was 15.3 months (95% CI, 9.8–20.9 months). In 19 censored patients, the median follow-up duration was 19.8 months (IQR, 9.2–31.2 months; range, 6.0–53.8 months).

In univariable Cox regression, a higher number of previous chemotherapy lines (*p* = 0.006), the presence of liver metastases (*p* = 0.024), the presence of brain metastases (*p* < 0.001), a higher tumor load on baseline PSMA-PET (*p* < 0.001), a higher PSA level prior to the first RLT cycle (*p* = 0.001), a lower red blood cell count prior to RLT (*p* < 0.001), lower hemoglobin (*p* = 0.003), a higher NLR (*p* = 0.011), and a higher De Ritis ratio before the first RLT cycle (*p* < 0.001) were associated with shorter OS (Table 2).

The number of previous chemotherapy lines (0 vs. 1 vs. 2), the presence of liver metastases (yes vs. no), presence of brain metastases (yes vs. no), tumor load on PSMA-PET (5 categories), the baseline PSA level (continuous; µg/L), red blood cell count (continuous; /pl), hemoglobin (continuous; g/dL), and the De Ritis ratio (continuous) were included in the stepwise inclusion process for multivariable Cox regression.

Following stepwise inclusion, only the number of previous chemotherapy lines (*p* = 0.036), the presence of brain metastases before the first RLT cycle (*p* < 0.001), the PSA level before the first RLT cycle (*p* = 0.004), and the De Ritis ratio before the first RLT cycle (*p* = 0.023) remained in the final multivariable Cox model (Table 3). This model showed a Harrell’s C (C index) of 0.71. A nomogram was built for this final multivariable Cox model to predict the OS of individual patients. The predicted survival probabilities matched well with the observed probabilities (Figure 3).

### 3.3. Simplified Risk Score to Predict Overall Survival

After the binarization of the variables PSA (>192 vs. ≤192 µg/L) and De Ritis ratio (>1.433 vs. ≤1.433; Figure 4), a prognostic score was built to illustrate the survival times of different risk groups that included all four variables from the final Cox model and contained five different risk groups.

This score allowed the five groups to differentiate with decreasing median overall survival and an increasing risk score (Table 4 and Figure 5; log-rank test, *p* < 0.001).

## 4. Discussion

In the current analysis, we identified or confirmed several factors that were predictors of shorter OS in patients with prostate cancer undergoing RLT with [^177^Lu]Lu-PSMA-617. Among these parameters, the De Ritis ratio showed an HR of approx. 1.3 per unit increase. This is slightly lower than the HR of 1.7 reported in a meta-analysis by Wu et al. that covered 18 studies and 9400 patients with different solid tumors [18]. However, the optimal cutoff observed in our analysis (De Ritis ratio > 1.433) is similar to the optimal cutoff reported by Miyake et al., who found that a high De Ritis ratio > 1.35 was associated with significantly shorter OS in patients with mCRPC undergoing Cabazitaxel chemotherapy [10]. Furthermore, Figure 4 (right) underlines that a broad range of possible cutoff values for the De Ritis ratio would have separated patients with significantly different survival rates, which underscores the robustness of its prognostic relevance.

Despite the systematic evidence of its prognostic value for OS, the causal mechanism remains to be fully understood. The De Ritis ratio is calculated as the ratio of AST and ALT. In cases of liver damage, the De Ritis ratio can reflect the time course and aggressiveness of liver disease (due to the different half-life of AST and ALT). However, stark increases in the De Ritis ratio (>1.5) are uncommon, apart from acute hepatitis, alcoholic liver disease or liver cirrhosis [19]. It is, therefore, hypothesized that the prognostic value of the De Ritis ratio in cancer may be primarily related to anaerobic glycolysis, which is typically increased in cancer cells. In this process, AST is required to provide oxaloacetate for the citric acid cycle [20]. Thornburg et al. showed that breast cancer cells are especially dependent on AST for a high proliferation rate [21]. Costello and Franklin reported that prostate cancer cells are characterized by increased citrate oxidation and a functional citric acid cycle [22]. Shao et al. investigated gene expression in prostate cancer cells with a Gleason score of >7 compared to normal prostate cells and cancer cells with a Gleason score of ≤7. They found the significant upregulation of several genes involved in the citric acid cycle (including the GOT2 gene, which encodes the mitochondrial form of AST) in prostate cancer cells with a Gleason score > 7 compared to other cell types [23]. In summary, AST elevation and its association with tumor aggressiveness might explain the prognostic value of the De Ritis ratio in patients with prostate cancer. Underlining this assumption, we found that patients with a high De Ritis ratio showed significantly higher AST (and lower ALT) than patients with a low De Ritis ratio (Table 1).

We also found a weak correlation between the De Ritis ratio and the PSA level, as well as the tumor load, when estimated from the baseline PSMA-PET. However, although this seems plausible, it does not prove that there is a causal relationship between the amount of tumor cells and the De Ritis ratio. It is also worth noting that both parameters, the De Ritis ratio and PSA, were independent prognostic factors in multivariable Cox regression, which argues against a simple correlation of both variables.

Another question that would require additional longitudinal analyses is how variable the De Ritis ratio is over time with respect to the change in tumor load during treatments, as well as the possible evolution of tumor biology and tumor aggressiveness.

Our analysis also confirmed several previous reports that low hemoglobin is associated with shorter OS after RLT [3,4,5,7], although it did not remain significant during variable selection for multivariable Cox regression. Furthermore, patients with liver metastases showed a shorter OS in previous studies [4,5,7,8], although this parameter also did not retain significance in multivariable Cox regression in the present analysis. In both variables (hemoglobin and liver metastases), the current analysis may be underpowered to assert statistical significance in the presence of other prognostic factors. In line with the present analysis, it has been reported that survival is prolonged in chemotherapy-naïve patients undergoing RLT [6,7,8]. In a study by Meyrick et al., patients with higher levels of PSA before RLT showed shorter survival [6].

The combined risk score in the current analysis achieved a Harrell’s C (C index) of 0.71 in the prediction of OS after RLT, which is comparable to a nomogram proposed and validated by Gafita et al. based on multicentric retrospective data (C index, 0.71) [7].

Some authors have investigated intratherapeutic parameters as prognostic factors for OS after RLT [3,5]. We restricted our prognostic analysis to factors available before RLT, which is a prerequisite if such a risk score is intended for prospective pretherapeutic survival prediction and, possibly, patient selection.

Our analysis is limited by the monocentric nature and lack of an independent patient sample to validate the prognostic value of single variables and the proposed risk score. The presence of brain metastases remained a predictor of OS in multivariable Cox regression. However, because brain metastases were infrequent (four patients in total), this result was constrained by a broad confidence interval and, thus, was not robust.

Furthermore, some other known prognostic factors could not be analyzed in the current cohort because they were not available (e.g., performance status [8], lactate dehydrogenase or C-reactive protein [4]). In our institution, [^18^F]FDG PET is only rarely performed in addition to PSMA PET prior to [^177^Lu]Lu-PSMA RLT due to reimbursement issues. It has been shown that patients with a mismatch between uptake in [^18^F]FDG PET and PSMA PET have worse prognosis [24]. However, there is currently no general requirement to perform an [^18^F]FDG PET prior to [^177^Lu]Lu-PSMA RLT [25], and Seifert et al. suggested that less than 5% of patients that are candidates for RLT show a FDG/PSMA mismatch that is only detected with the additional [^18^F]FDG PET [26].

It is worth noting that the current analysis examined the prognostic relevance of parameters, not their predictive value, meaning that the prognostic score does not help to determine if patients might benefit from RLT or different treatments. We can, therefore, not conclude that patients with a high risk of unfavorable OS according to the proposed model should receive a certain different therapy. However, if a particular patient group exhibits a notably short OS, they may be more aptly suited for best supportive care (or possibly other systemic therapies) rather than RLT.

## 5. Conclusions

Our analysis adds to the body of evidence, which shows that a high De Ritis ratio predicts shorter OS in different tumor types, including prostate cancer, and the context of RLT with [^177^Lu]Lu-PSMA-617. As part of the proposed combined risk score, it could help to better predict individual survival on or after RLT. Although the current analysis did not investigate the predictive value of this score (i.e., its capability to identify patients that respond to a certain treatment especially well), it might help to identify those patients with an especially unfavorable survival outlook. In these patients, the risks and benefits of RLT could be weighed even more thoroughly.

However, an independent external validation of both these results and the proposed risk score is required.

## Figures and Tables

**Figure 1 cancers-15-04907-f001:**
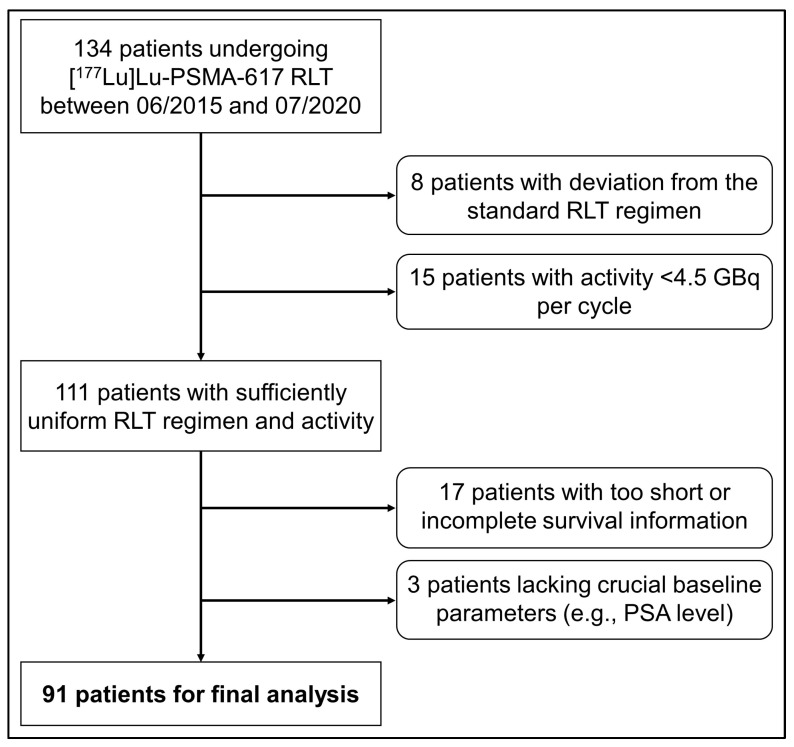
Flow diagram of patient inclusion.

**Figure 2 cancers-15-04907-f002:**
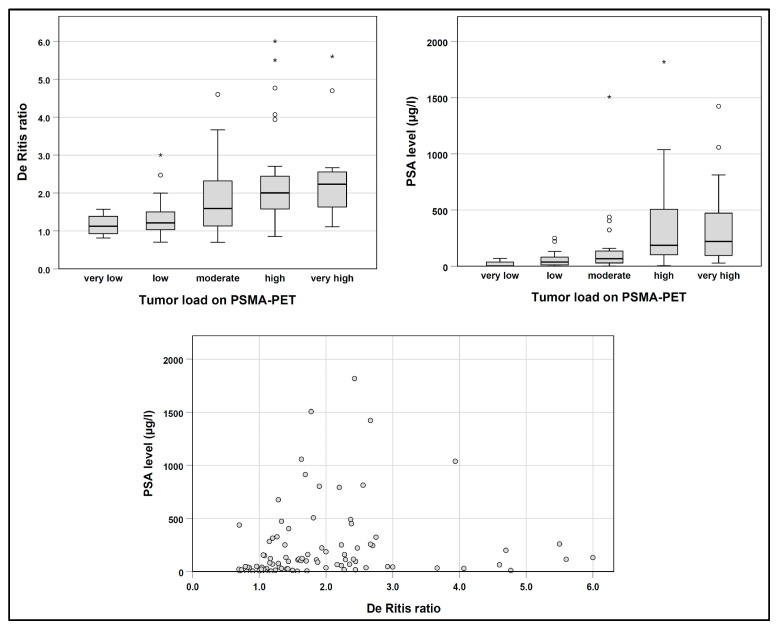
Association between baseline De Ritis ratio, PSA and tumor load on PSMA-PET. Extreme values are marked with *.

**Figure 3 cancers-15-04907-f003:**
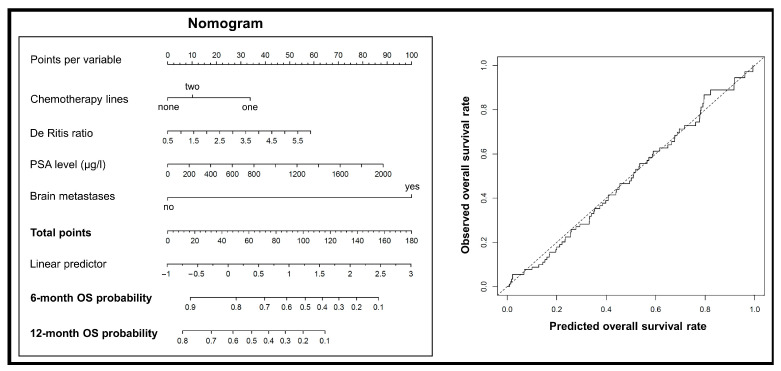
(**left**) A nomogram based on the multivariable Cox model to predict OS of patients. (**right**) A plot comparing the predicted OS using the multivariable model with observed OS.

**Figure 4 cancers-15-04907-f004:**
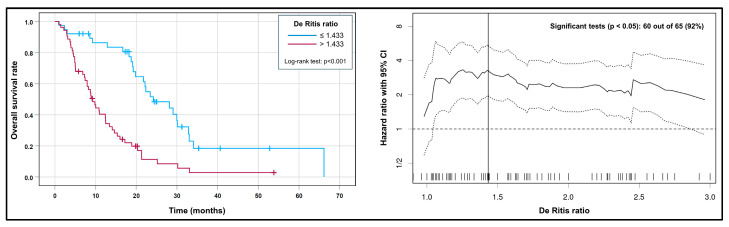
(**left**) Kaplan–Meier curves for the high vs. low De Ritis ratio before the first RLT cycle. (**right**) A plot showing the HR (solid line) and its 95% CI (dashed lines) at various possible thresholds for the De Ritis ratio. Out of 65 possible thresholds, 60 (92%) resulted in a significant log-rank test (i.e., the lower bound of the 95% CI lies above the reference line of HR = 1), which shows that the prognostic value of the De Ritis ratio is relatively robust against possible cutoff values. The optimal threshold identified by the lowest *p*-value in the log-rank test (>1.433) is marked with a vertical line.

**Figure 5 cancers-15-04907-f005:**
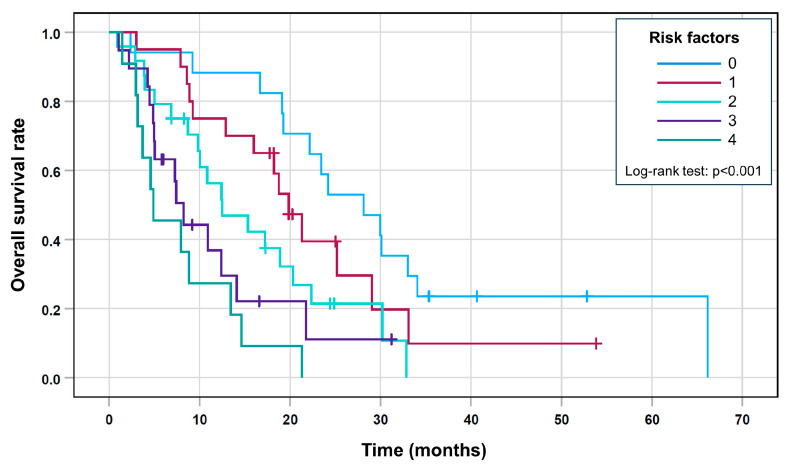
Kaplan–Meier curves for the risk score.

**Table 1 cancers-15-04907-t001:** Patient characteristics (patient count or median (IQR)) and comparison of patients with high (>1.433) vs. low De Ritis ratio (≤1.433) before the first RLT cycle. Significant *p*-values are printed in bold (Wilcoxon rank-sum or Fisher’s exact test).

Variables	Total	High De Ritis	Low De Ritis	*p*-Value
Patient count	91 (100%)	53 (58%)	38 (42%)	
Age (years)	70 (65–76)	72 (65–77)	69 (64–74)	0.3
Number of RLT cycles				0.25
1	9 (10%	6 (11%)	3 (8%)	
2	31 (34%)	22 (42%)	9 (24%)	
3	32 (35%)	16 (30%)	16 (42%)	
4	17 (19%)	8 (15%)	9 (24%)	
5	1 (1%)	1 (2%)	0	
6	1 (1%)	0	1 (3%)	
Average activity per cycle (GBq)	6.0 (5.7–6.0)	6.0 (5.5–6.0)	6.0 (5.8–6.0)	0.29
Cumulative activity (GBq)	17.3 (11.9–18.2)	12.2 (11.1–18.1)	17.9 (12.1–23.3)	0.06
**Previous treatments**				
Prostatectomy	51 (56%)	30 (57%)	21 (55%)	1.0
Androgen deprivation	91 (100%)	53 (100%)	38 (100%)	n.a.
Radiotherapy (any)	64 (70%)	39 (74%)	25 (66%)	0.49
Abiraterone	69 (76%)	43 (81%)	26 (68%)	0.22
Enzalutamide	60 (66%)	36 (68%)	24 (63%)	0.66
Abiraterone or Enzalutamide	82 (90%)	49 (92%)	33 (87%)	0.19
Radium-223-dichloride	19 (21%)	14 (26%)	5 (13%)	0.31
Previous chemotherapy lines				**0.014**
None	33 (36%)	13 (25%)	20 (53%)	
One line	32 (35%)	24 (45%)	8 (21%)	
Two lines	26 (29%)	16 (30%)	10 (26%)	
**Pattern of metastases**				
Lymph nodes	79 (87%)	44 (83%)	35 (92%)	0.35
Bone	87 (96%)	52 (98%)	35 (92%)	0.3
Liver	17 (19%)	14 (26%)	3 (8%)	**0.03**
Lung	10 (11%)	7 (13%)	3 (8%)	0.51
Brain	4 (4%)	3 (6%)	1 (3%)	0.64
**Tumor load on PSMA-PET**				**<0.001**
Very low	4 (4%)	1 (2%)	3 (8%)	
Low	22 (24%)	6 (11%)	16 (42%)	
Moderate	23 (25%)	13 (25%)	10 (26%)	
High	25 (28%)	19 (36%)	6 (16%)	
Very high	17 (19%)	14 (26%)	3 (8%)	
**Laboratory parameters**				
PSA before the first RLT cycle (µg/L)	95 (28–250)	117 (59–364)	37.6 (12.2–137)	**<0.001**
Alkaline phosphatase (U/L) ^1^	118 (77–189)	131 (89–234)	92 (72–141)	0.06
eGFR (CKD-EPI)	88 (74–90)	89 (76–90)	87 (71–90)	0.63
White blood cell count (/nL)	6.2 (4.9–7.3)	6.2 (4.7–7.2)	6.3 (5.3–7.8)	0.4
Red blood cell count (/pl)	4.1 (3.6–4.4)	3.9 (3.5–4.2)	4.4 (4.0–4.5)	**<0.001**
Hemoglobin (g/dL)	12.1 (10.5–13.1)	11.4 (9.7–12.5)	12.7 (11.4–13.4)	**<0.001**
Platelet count (/nL)	246 (191–287)	246 (197–305)	248 (187–270)	0.58
Neutrophil–lymphocyte ratio ^2^	4.4 (3.1–7.2)	5.2 (3.8–7.6)	3.3 (2.6–5.6)	**0.01**
Platelet–neutrophil ratio ^2^	56.9 (43.1–80.8)	58.0 (44.9–84.0)	55.3 (42.1–77.5)	0.43
De Ritis ratio	1.63 (1.16–2.38)	2.28 (1.75–2.68)	1.1 (0.95–1.27)	**<0.001**
Aspartate transaminase (U/L)	28 (23–41)	38 (26–50)	24 (20–29)	**<0.001**
Alanine transaminase (U/L)	18 (14–25)	15 (13–19)	23 (18–29)	**<0.001**
Death during follow-up	72 (79%)	47 (89%)	25 (66%)	**0.01**

^1^ Missing values in 16 patients. ^2^ Missing values in 14 patients.

**Table 2 cancers-15-04907-t002:** Univariable Cox regression. Significant *p*-values are printed in bold.

Variables	HR	95% CI	*p*-Value
Age (years)	0.99	0.96–1.02	0.61
**Previous treatments**			
Prostatectomy	0.73	0.46–1.16	0.18
Androgen deprivation	n.a. ^1^		
Radiotherapy (any)	0.95	0.57–1.56	0.83
Abiraterone	1.26	0.74–2.16	0.4
Enzalutamide	0.96	0.59–1.56	0.88
Radium-223-dichloride	1.27	0.74–2.17	0.39
Previous chemotherapy lines			**0.006**
None	reference		
One line	2.53	1.43–4.47	**0.001**
Two lines	1.77	0.93–3.37	0.085
**Pattern of metastases**			
Lymph nodes	0.62	0.33–1.15	0.13
Bone	1.26	0.4–4.02	0.7
Liver	1.98	1.09–3.59	**0.024**
Lung	0.89	0.41–1.94	0.77
Brain	6.63	2.26–19.44	**<0.001**
**Tumor load on PSMA-PET ^2^**	1.37	1.13–1.67	**<0.001**
**Laboratory parameters**			
PSA before first RLT cycle (µg/L)	1.001	1.0–1.001	**0.001**
Alkaline phosphatase (U/L) ^3^	1.001	1.0–1.001	0.076
eGFR (CKD-EPI)	1.0	0.98–1.01	0.69
White blood cell count (/nL)	0.94	0.82–1.08	0.36
Red blood cell count (/pl)	0.46	0.3–0.71	**<0.001**
Hemoglobin (g/dL)	0.81	0.7–0.93	**0.003**
Platelet count (/nL)	1.0	1.0–1.0	0.89
Neutrophil–lymphocyte ratio ^4^	1.03	1.01–1.05	**0.011**
Platelet–neutrophil ratio ^4^	1.001	0.99–1.01	0.81
De Ritis ratio	1.4	1.17–1.68	**<0.001**

^1^ All patients previously had androgen deprivation therapy. ^2^ Categories 1–5 (“very low” to “very high”). ^3^ Missing values in 16 patients. ^4^ Missing values in 14 patients.

**Table 3 cancers-15-04907-t003:** The final multivariable Cox model. Significant *p*-values are printed in bold.

Variables	HR	95% CI	*p*-Value
Previous chemotherapy lines			**0.036**
None	reference		
One line	2.12	1.18–3.83	**0.012**
Two lines	1.25	0.6–2.6	0.55
Brain metastases: yes	9.2	3.03–27.9	**<0.001**
PSA (µg/L; continuous)	1.001	1.000–1.002	**0.004**
De Ritis ratio (continuous)	1.27	1.03–1.56	**0.023**

**Table 4 cancers-15-04907-t004:** Survival estimates for each group with risk scores (Kaplan–Meier method).

Number of Risk Factors	Overall Survival (Months)	
	Median	95% CI
0	28.1	19.4–36.9
1	19.8	16.3–23.4
2	12.5	5.8–19.1
3	8.2	6.5–10.0
4	4.9	0.3–9.4

## Data Availability

The data that are required to reproduce the reported results (predictor variables and survival information for each patient) are available in anonymized form at the Zenodo Open Data repository (URL/DOI: https://doi.org/10.5281/zenodo.8262551 (accessed on 28 September 2023)).

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
