# Peer review of "Prognostic Value of the De Ritis Ratio for Overall Survival in Patients with Metastatic Castration-Resistant Prostate Cancer Undergoing [177Lu]Lu-PSMA-617 Radioligand Therapy"

_cancers, 2023, doi:10.3390/cancers15204907_

Round 1

Reviewer 1 Report

The authors showed De Ritis ratio had prognostic value for classify overall survival of prostate cancer patients who received Lu-PSMA radioligand therapy.

The reviewer thinks that the article is acceptable basically for “Cancers”, but there are several requirements before accepting the article.

Ø  In table 3 “Brain metastasis” should be “Liver metastasis”.  Please check it.

Ø  Why did the authors select De Ritis ratio?  There are a lot of serum biochemical factors that may related with the patient’s prognosis, such as NL ratio or Platelet to neutrophil ratio.

Ø  The reviewer thinks that it is required to weighing for each factor for risk classification, since HR might be different among each factor.

Ø  mCRPC patients have heterogenous disease characters.  They might posse PSMA-PET negative but FDG-PET positive lesions.  How did the authors characterize and treat these patients with [177Lu]-PSMA-617 radioligand therapy.

Ø  How did the authors use their own prognostic risk classification?  Who are the targets of [177Lu]-PSMA-617 radioligand therapy.  On the other hand, what kind of treatment do you select for those who may not be a candidate of [177Lu]-PSMA-617 radioligand therapy?

Ø  The authors should present radiological factors that may be a candidate of [177Lu]-PSMA-617 radioligand therapy.  The authors might be able to calculate sum of SUV by [68Ga]Ga-PSMA-11.

Author Response

Please see the attached document with our responses.

Reviewer 2 Report

This is a well-written original retrospective study determining that the AST/ALT (De Ritis) ratio has prognostic value for patients receiving Lu-177-PSMA-617. As this is a recent regulatory approval drug, there are still several unknowns as to what factors contribute to its success and disappointment. This is one study that tries to shed some light onto this issue.

The biggest improvement that can be made is explaining why the cutoff was 1.433. This is only briefly touched on in the discussion. Different de Ritis cutoffs can be included in the introduction, and in the methods or results explain how 1.433 was determined to be the best one - and not 1.4 or 1.5 for example.

Line 212: should say "in cases of"

Author Response

(The authors gave the same response as above.)

Reviewer 3 Report

The study is very interesting and well-presented. I am more of a novice in statistical calculations than the authors but found I could easily follow the reasoning. I have no real criticism of the paper, possibly a few questions:

In the final model presented in Table 3, the only non-significant variable is “Two lines of chemotherapy”, while one line is significant. Would it not have made more sense to have “Chemotherapy (any)” as the variable, or was it just that the patients had one, not two lines of chemotherapy that was predictive of survival?

In the discussion it seems that the increase in AST would be related to the number of cancer cells in the body, and so would of course the PSA. So is not the model in some way saying the rather obvious, i.e. higher load of disease, worse prognosis?

Do the authors suspect that the De Ritis ratio would correlate with any parameters of PSMA PET/CT images of the patients, total lesion volume or similar?

Does the De Ritis ratio change significantly during disease progression and/or therapy? Is “before the first cycle of RLT” a similar enough situation for the different patients or could the ratio be influenced by time since last anti-androgen or chemotherapy?

Some of these questions might merit some further discussion in the paper?

Author Response

(The authors gave the same response as above.)

Round 2

Reviewer 1 Report

The authors answered most of my questions.

It is ready to accept for the journal.